# TEST-TIME ADAPTATION FOR LLM AGENTS VIA ENVIRONMENT INTERACTION

**Arthur Chen**[1]* **Zuxin Liu**[2] **Jianguo Zhang**[2] **Akshara Prabhakar**[2] **Zhiwei Liu**[2]
**Shelby Heinecke**[2] **Silvio Savarese**[2] **Victor Zhong**[1]† **Caiming Xiong**[2]†

[1]University of Waterloo     [2]Salesforce AI Research

## ABSTRACT

Large language model (LLM)-based agents struggle to generalize to novel and complex environments, such as unseen websites or new sets of functions, due to a fundamental mismatch between their pre-training and test-time conditions. This challenge stems from two distinct failure modes: a syntactic misunderstanding of environment-specific components like observation formats, and a semantic misunderstanding of state-transition dynamics, which are only revealed at test time. To address these issues, we propose two distinct strategies for adapting LLM agents by leveraging environment-specific information from interaction that is available during deployment. First, an *online syntactic alignment* (SA) method parameterizes environmental nuances by learning a lightweight adaptation vector that biases the model's output distribution, enabling rapid alignment with an environment response format. Second, a *deployment-time dynamics grounding* (DG) method employs a persona-driven exploration phase to systematically probe and learn the environment's causal dynamics before task execution, equipping the agent with an in-context world model. We evaluate these strategies across diverse agentic benchmarks, including function calling and web navigation. Our empirical results show the effectiveness of both strategies across all benchmarks with minimal computational cost. We find that dynamics grounding is particularly effective in complex environments where unpredictable dynamics pose a major obstacle, demonstrating a robust path toward more generalizable and capable LLM-based agents. For example, on the WebArena multi-site split, this method increases the agent's success rate from 2% to 23%. We release our code[1].

## 1  INTRODUCTION

Large language model (LLM)-based AI agents have demonstrated remarkable capabilities across diverse tasks (Li et al., 2023a; Zhou et al., 2024a; Patil et al.; Xie et al., 2024), yet they often fail to generalize when deployed in novel agentic environments such as web navigation (Zhou et al., 2024a; He et al., 2024) and function calling (Patil et al.; Yao et al., 2024). These failures reflect two core challenges—the systematic mismatch between the agent's pre-trained knowledge and the specific syntax and dynamics of the deployed environment (Yang et al., 2025; Thil et al., 2024). Although both mismatches stem from a lack of prior knowledge, we distinguish them operationally because they require different adaptation mechanisms (syntactic vs. semantic). To illustrate these two failure patterns, we consider an LLM agent tasked with booking a flight on a previously unseen travel website:

1. **Syntactic Mismatch**: the LLM agent's prior knowledge does not align with environment-specific information, such as observation structure (Yang et al., 2024; Gur et al., 2023) and the environment's unique syntax (Lei et al., 2024; Chen et al., 2024). This causes parsing and context-understanding issues: a pre-trained LLM agent might try `click(``Search'')` or target

---

*Work done during internship. Correspondence to: `haonan.chen@uwaterloo.ca`
†Equal advising.
[1]`https://github.com/r2llab/GTTA`

    `destination`, while the new site exposes `Go` and `dest_field`, producing invalid actions and immediate failures. (Yang et al., 2024; Gur et al., 2023; Lei et al., 2024; Chen et al., 2024)

2. **Semantic Mismatch**: the agent lacks an accurate, environment-specific causal model of state transitions, so it cannot predict the consequences of actions. For example, the agent may expect `click(``Go'')` to show flight results, but the site instead opens a date-confirmation pop-up; without knowing this transition, the agent cannot form the correct multi-step plan and fails. (Forrester, 1971; Chae et al., 2025; Zhang et al., 2024)

Current approaches to this generalization gap are ill-suited for rapid adaptation in novel environments. In addition, many methods require human-annotated or LLM-annotated demonstrations (Wang et al., 2024b; Murty et al., 2025; Xu et al., 2024), which can be resource-intensive or rely on LLM's prior knowledge about the environment. On the other hand, to address semantic mismatch, explicit world modeling like (Chae et al., 2025) involves a heavyweight pipeline of extensive data collection and fine-tuning a separate, specialized model that is computationally expensive and struggles to generalize without being retrained. Both paradigms present significant overhead, highlighting the need for more efficient strategies that can ground an agent using only the information available at test time.

To address the two failure modes and close the generalization gap, we propose two annotation-free strategies that operate at deployment time. (1) Online syntactic alignment (SA)—a lightweight module that learns a small per-interaction bias (an adaptation vector applied to late features) to quickly align an agent's output distribution with environment-specific syntax. Observing `Go` and `dest_field` on the page, for example, lets the adapter steer action generation toward the site's actual element names. (2) Dynamics grounding (DG)—a short, persona-guided exploration phase that iteratively collects in-context rules about state transitions (e.g., clicking `Go` opens a date pop-up) and supplies those rules as context for downstream planning. Both methods require only test-time observations and no trajectory annotations or expensive fine-tuning.

We evaluate our methods on function-calling and web-navigation benchmarks. Both approaches yield consistent improvements: syntactic alignment gives efficient, steady gains across sites, while dynamics grounding produces particularly large improvements on environments with unfamiliar transitions. For example, on the WebArena multi-site split (Zhou et al., 2024a), dynamics grounding raises `GPT-4.1` success from 2% to 23%. Taken together, syntactic and semantic test-time grounding substantially narrow the deployment generalization gap and offer a practical path toward more robust LLM agents.

## 2   Related Work

**Test-Time Adaptation**    Originating in computer vision, test-time adaptation (TTA) addresses distributional shifts between training and testing data by adapting models at inference time (Wang et al., 2021; Niu et al., 2022; Sun et al., 2020). Our syntactic alignment builds directly on methods that update model parameters or steering vectors at test time to minimize an unsupervised objective, such as entropy minimization or self-supervised learning (Wang et al., 2021; Hu et al., 2025; Li et al., 2023b). Our syntactic alignment aligns with steering vector research (Tan et al., 2025; Sinii et al., 2025; Panickssery et al., 2024; Turner et al., 2024) in its use of latent space biases to shift output distributions. However, standard steering approaches generally apply modulation vectors to shift high-level traits (Tan et al., 2025), such as honesty (Zou et al., 2025). We distinguish our work by tailoring this mechanism to the dynamic nature of agentic tasks: our vector is treated as a temporary parameter, updated online using the environment's response as a self-supervisory signal, and reset per episode to ensure precise, context-dependent alignment without interference across tasks. While TTA for LLMs is emerging for tasks like math reasoning (Zuo et al., 2025) and few-shot learning (Akyürek et al., 2025), its application to the unique, interactive, and stateful challenges of LLM agents remains underexplored. Our work is the first to systematically apply this paradigm to adapt agents to novel environment formats without supervised trajectories.

**Environment Modeling for Agents**    Recent work has focused on incorporating environment dynamics into LLM-based planning by training a parameterized world model (Fang et al., 2025; Chae et al., 2025; Ding et al., 2026; Qiao et al., 2024), or using an LLM to predict the next state using continuously updated environment rules (Zhou et al., 2024b). Such approaches typically require

collecting a large set of observations for learning a world model, and can be computationally expensive (Chae et al., 2025). In contrast, our dynamics grounding introduces a lightweight, deployment-time pipeline to generate a temporary, in-context world model from a few exploratory interactions. This aligns with a broader trend of using in-context learning for adaptation, including methods like agent workflow memory (Wang et al., 2024b) and retrieval-augmented generation (RAG) for planning (Luo et al., 2023; Kagaya et al., 2024; Wang et al., 2024a). However, our contribution lies in the systematic and automated process of generating this knowledge via persona-driven exploration, removing the need for pre-existing annotated demonstrations or a separate, trained dynamics model.

**LLM-Based Agents** Large language models (LLMs) have become the planning backbone of a wide range of agentic systems due to their strong instruction-following capabilities and generalization across diverse tasks (Ouyang et al., 2022; Mishra et al., 2022; Zhou et al., 2024a). Their ability to interpret complex instructions and generate coherent action plans has enabled their integration into applications such as web navigation (Zhou et al., 2024a; He et al., 2024; Lù et al., 2024; Deng et al., 2023) and function calling (Patil et al.; Yao et al., 2024). Despite recent advances, LLM agents struggle to comprehend the specific syntactical input structure of different environments, such as in web navigation (Yang et al., 2024; Gur et al., 2023) and text-to-SQL (Lei et al., 2024). This is due to the fundamental mismatch between the LLM pre-training corpus and the specific agentic environment (Chen et al., 2024; Ben-David et al., 2010).

## 3 Grounded Test-Time Adaptation Strategies

In this section, we detail our two adaptation strategies which mitigate syntactic and semantic gaps. We first formalize the problem setup, which is designed to reflect realistic deployment scenarios and is defined by three key constraints:

1. **No annotated trajectories or offline data.** The agent cannot access expert demonstrations or pre-collected data from the target environment. It must learn from scratch during interaction. For our running travel example, the agent has never seen travel-example.com and has no pre-existing dataset of successful bookings on that site.

2. **Online, streaming adaptation.** The agent receives tasks one at a time and must adapt on-the-fly without seeing a full batch of test instances in advance. For example, it must adapt to the website's quirks while handling a single request (e.g. "Book a flight from New York to Saskatoon") before seeing any future tasks.

3. **Permitted test-time interaction.** The agent is allowed to interact with the environment to gather information before receiving a specific task, but without supervision. For instance, it can perform a "blind" exploration of travel-example.com to discover that a button triggers a calendar pop-up, but it does not know what kind of tasks the user will ultimately ask it to execute.

This setting mirrors constraints in practical benchmarks (e.g., BIRD (Li et al., 2023a), WebArena (Zhou et al., 2024a)) and necessitates methods that can leverage only unsupervised, test-time interaction. To formulate this challenge, at test-time the LLM input $\mathcal{I}$ is constructed as

$$\mathcal{I} = [p; o; \{a\}_{i=1}^{T-1}] \tag{1}$$

where $p$ denotes the task instruction (e.g., rules, task description), $o$ denotes the current environment observation (e.g., accessibility tree observation, response of function call), and $\{a\}_{i=1}^{T-1}$ denotes the sequence of past actions taken by the LLM-based agent up to the current step. For environments with short or simple observations (e.g., tool-use scenarios), we instead construct the model input using all previous interactions between the agent and the environment, which may include both environment responses and agent actions. With this formulation of the challenge in mind, we describe two adaptation methods in the remainder of this section.

### 3.1 Syntactic Alignment

To address syntactic mismatches that arise in novel environments, we propose *syntactic alignment*. The objective is to parametrically adapt the model's output distribution at test time by treating the current context (i.e. task instructions, observations, and action history) as a self-supervisory signal.

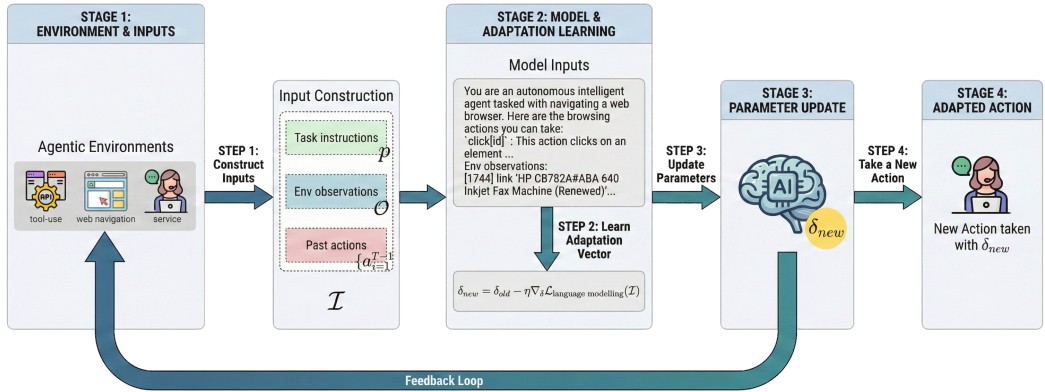

Figure 1: **Overview of syntactic alignment (SA).** This figure includes an example of web navigation shopping task to illustrate how the agent adapts to new environment. (1) At the start of each episode, we initialize an adaptation vector $\delta$ as a zero vector and construct inputs to the LLM agent. (2) During task execution, the agent receives environment instructions and observations. (3) At each step, we update the adaptation vector using cross-entropy loss on the current input, and apply the adaptation vector as a bias to the LLM's final hidden layer. This enables rapid alignment to environment-specific observation and action formats. (4) The LLM agent takes a new action with the updated vector, which shifts the model's output distribution to better match the test-time environment.

This allows the model to align with local syntactic patterns, such as unique UI element labels or response formats, without requiring a deep understanding of the environment's causal dynamics.

Our approach introduces a single lightweight adaptation vector $\delta \in \mathbb{R}^d$, where $d$ is the hidden dimension of the language model. This vector acts as an additive bias to the final hidden representations $H \in \mathbb{R}^{n \times d}$ before the final projection layer. At each step of an episode, the adapted logits are computed as:

$$\text{logits}' = (H + \delta)W_{\text{LM}}^T \tag{2}$$

where $W_{\text{LM}} \in \mathbb{R}^{|V| \times d}$ is the model's output projection matrix and $|V|$ is the vocabulary size.

This vector $\delta$ is updated at each turn by performing one gradient descent step to minimize the language modeling loss of the current input context $\mathcal{I}$ according to the following:

$$\delta_{\text{new}} \leftarrow \delta_{\text{old}} - \eta \nabla_\delta \mathcal{L}_{\text{CE}} \left( f_{\theta,\delta}(\mathcal{I}_{1:n-1}), \mathcal{I}_{2:n} \right) \tag{3}$$

where $f_{\theta,\delta}$ is the LLM parameterized by its fixed weights $\theta$ and the adaptable vector $\delta$, and $\eta$ is the learning rate. $\mathcal{I}_{1:n-1}$ denotes the input subsequence consisting of tokens 1 through $n - 1$, and $\mathcal{I}_{2:n}$ denotes the corresponding next-token targets obtained by shifting the sequence by one position. The loss in Eq. 3 is therefore computed by predicting each token in $\mathcal{I}_{2:n}$ from its preceding context $\mathcal{I}_{1:n-1}$. This update encourages the model to internalize the syntactic features present in the immediate context by adapting its output distribution to the environment context.

To illustrate this in practice, consider our agent on the unseen website `travel-example.com`. The agent's input context $\mathcal{I}$ contains the accessibility tree, which includes strings such as `[145] <button>Go` and `dest_field`.

1. **Initialization**: At the episode's start, the adaptation vector is initialized as a zero vector, $\delta \leftarrow 0$.

2. **Prediction**: Without adaptation, the model's logits would likely favor the action `click(``Search'')`, based on its pre-trained priors.

3. **Adaptation**: We compute the language modeling loss on the context $\mathcal{I}$ (Eq. 3). Because the strings `Go` and `dest_field` are present in $\mathcal{I}$, the gradient update to $\delta$ will shift its values to favor their tokens.

4. **Correction**: In the subsequent generation step, the adapted logits (Eq. 2) now assign a higher probability to the syntactically correct action, `click(``Go'')`.

This process is computationally efficient, and steps 2-4 are repeated at every model prediction within an episode. To prevent catastrophic forgetting and ensure that adaptation is specific to the current

environment, $\delta$ is reset to zero at the beginning of each new episode. This lightweight update aligns the model's pre-trained distribution with the specific distribution of the test-time environment.

## 3.2 DYNAMICS GROUNDING

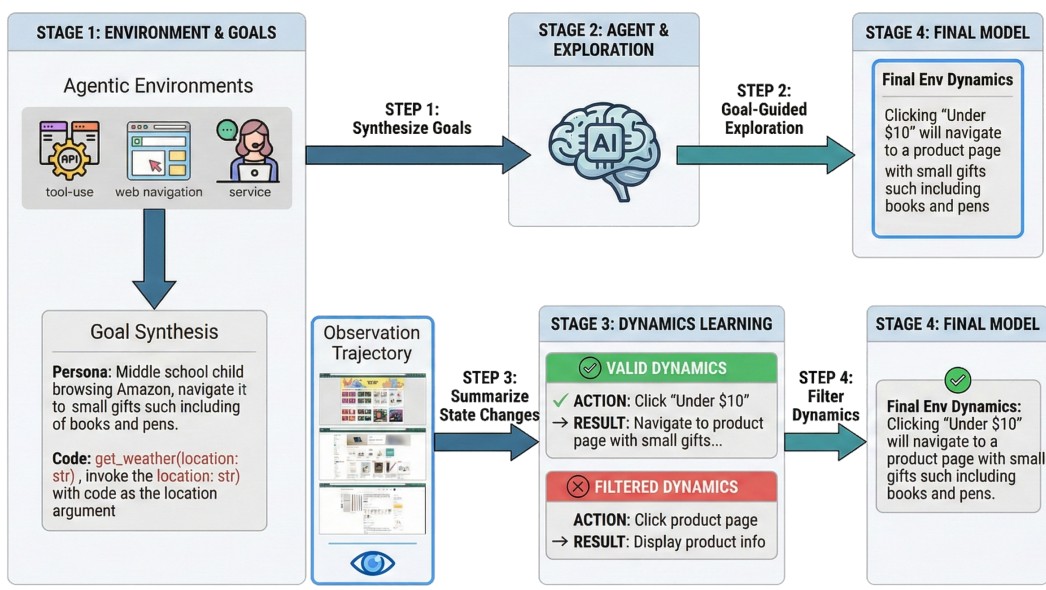

Figure 2: **Overview of dynamics grounding (DG).** This figure includes an example of web navigation shopping task to illustrate how the pipeline generates environment dynamics in language. (1) We synthesize diverse exploration tasks based on personas using environment descriptions. (2) An exploration agent interacts with the environment to collect interaction logs of state transitions. (3) An LLM extracts and summarizes environment dynamics from these logs. (4) A reasoning model filters less informative rules, which are then used to augment the agent's context during evaluation, enabling more transition-aware decision making.

LLM-based agents often fail in novel environments due to a **semantic mismatch**: they lack a "world model" to predict the outcomes of their actions (Ha & Schmidhuber, 2018; Chae et al., 2025). To address this, we propose *dynamics grounding*, a strategy that proactively discovers the causal dynamics of an environment **before the agent encounters any user tasks during deployment**. The objective is to construct an in-context world model using natural language by performing a one-time, automated exploration. The cost of this discovery phase is amortized over all subsequent tasks performed in that environment.

Our method follows a four-step pipeline to generate a concise set of environment dynamics, denoted as $E_{\text{clean}}$, for a given new environment.

1. **Persona/Exploratory Goal Synthesis.** An LLM is prompted with a high-level description of the environment (e.g., website purpose, API documentation) to generate a set of $N$ diverse, exploratory "personas" or "goals". Personas frame open-ended tasks designed to probe for non-obvious interactions. For our `travel-example.com` scenario, a synthesized persona might be: *"As a first-time user, I want to see what happens if I try to search for a flight without selecting a date.".* We prefer personas over a less-structured exploration policy (e.g. maximizing state novelty or coverage) to guide the agent to explore complex, multi-step interactions that a naive exploration strategy might miss, thus yielding more semantically rich dynamics.

2. **Exploration and on-the-fly Dynamics Extraction** For each persona or goal, an LLM agent interacts with the environment. The agent is instructed to take novel actions to maximize its discovery of new state transitions. After each action is executed and the system transitions into the next state, we take (observation, action, new observation) to summarize into a concise human-readable rule $e$. Immediately after, the list of generated rules $\{e\}_{i=1}^{T}$ is appended to the agent to encourage taking actions that have not been taken before. In our example, the agent would execute `click(''Go'')` and log the transition from the main page to a state where a calendar

pop-up is active. Then the transition on `travel-example.com` would be extracted as the dynamic: Executing the action `click('Go')` on the main page causes a calendar modal to appear for date selection.

3. **Filtering and Consolidation.** The set of all extracted dynamics, $E = \{e\}_{i=1}^{M}$, is passed to a reasoning model. This model filters out trivial (e.g., "typing in a text field shows text") or repetitive rules, producing a final, clean set of dynamics, $E_{\text{clean}}$.

During task execution at test time, we augment the input context $\mathcal{I}$ with discovered dynamics:

$$\mathcal{I}' = [\mathcal{I}; E_{\text{clean}}] \tag{4}$$

By leveraging the LLM's in-context learning capabilities, this explicit knowledge of the environment's state-transition patterns guides the agent to make more informed and transition-aware decisions. In our example, with the calendar pop-up dynamic in its context, the agent correctly anticipates the outcome of its click and plans its subsequent actions accordingly, successfully navigating the semantic mismatch. Example prompts for exploration and dynamics extraction are shown in Appendix sections section A.2 and section A.3.

## 4 EXPERIMENTS

### 4.1 SETUP

We evaluate our approach on three realistic agentic benchmarks: BFCLv3 (Patil et al.), WebArena (Zhou et al., 2024a), and Tau-Bench (Yao et al., 2024), which together cover both web navigation and conversational function calling tasks. As for baselines, we evaluate open-source `Qwen2.5-14B-Instruct` (Qwen et al., 2025) model and closed-source `GPT-4.1`. On WebArena, we additionally compare to the World-model-augmented (WMA) agent from (Chae et al., 2025) as a baseline. This approach first collects exploration demonstrations, then trains a `Llama-3.1-8B-Instruct` (Grattafiori et al., 2024) model to predict the next state given an action (i.e., a learned world model). During evaluation, this world model is used to provide next-state predictions

Table 1: Number of tasks per website in the WebArena benchmark. The benchmark consists of six websites, including a multi-site category for tasks that require interacting across multiple websites (from the six sites), for a total of 812 tasks.

| Website | Number of Tasks |
|---|---|
| Shopping | 187 |
| Shopping Admin | 182 |
| GitLab | 180 |
| Map | 109 |
| Reddit | 106 |
| Multi-site | 48 |
| Total | 812 |

to the agent on-the-fly, helping the agent reason about the consequences of its actions. In this comparison, we use the same `GPT-4o mini` as the original WMA experiments. Aside from the syntactic alignment and dynamics grounding experiments, we provide ablation studies in section 4.3.

For all dynamics grounding experiments, we leverage `GPT-4.1` as an exploration policy to extract environmental dynamics. For each environment, we conduct 10 exploration episodes to gather a diverse set of dynamics. For environment dynamics filtering, we apply `o3`[2] to remove repetitive and trivial environment dynamics. We also include ablation studies on the number of exploration episodes and the effect of filtering in section 4.3.

Before delving into the results, we first introduce the benchmarks and describe the specific experimental setups used for experiments.

**WebArena** The WebArena benchmark consists of five self-hosted websites and a total of 812 tasks. Many tasks require interacting across multiple websites, which poses additional challenges for LLM-based agents. For evaluation, WebArena employs a string-matching-based mechanism to assess correctness, providing more stable and objective results compared to benchmarks using LLMs to score, such as (He et al., 2024; Gou et al., 2025). For dynamics grounding, we first synthesize 10 personas via prompt instruction 1 for each website using GPT-4o (OpenAI et al., 2024) and with seed personas from NNetNav (Murty et al., 2025). Then for each persona, we roll-out a `GPT-4.1` agent to explore the environment guided by the persona description, with the exploration prompt

---

[2]`https://platform.openai.com/docs/models/o3`

Table 2: Main results on WebArena, WebVoyager, BFCLv3, and Tau-Bench benchmarks. We report task success rates (%) for each model and adaptation method. For Tau-Bench, we average runs across 5 seeds and use a custom more stable codebase. Both test-time adaptation strategies improve performance. "N/A" indicates not applicable—AWM requires training a web-specific world model on state transitions, which is not applicable to Tau-Bench (conversational only) or BFCLv3 (no explicit web-like state transition data); Dynamics grounding does not operate in conversational Tau-Bench as there is no explicit and fixed state transition rules in the environment.

| Model | Task | | | |
|---|---|---|---|---|
| | WebArena | BFCLv3 | Tau-Bench | |
| | | | Airline | Retail |
| GPT-4.1 | 30.0 | 55.5 | - | - |
| GPT-4.1 (+ DG) | 35.0 (+5.0) | 64.0 (+8.5) | N/A | N/A |
| GPT-4o mini | 12.0 | - | - | - |
| GPT-4o mini (+ WMA) | 13.5 (+1.5) | N/A | N/A | N/A |
| GPT-4o mini (+ DG) | 18.0 (+6.0) | - | N/A | N/A |
| Qwen2.5-14B-Instruct | 17.0 | 18.5 | 21.6 | 43.3 |
| Qwen2.5-14B-Instruct (+ SA) | 18.0 (+1.0) | 20.0 (+1.5) | 25.2 (+3.6) | 44.9 (+1.6) |
| Qwen2.5-14B-Instruct (+ DG) | 20.0 (+3.0) | 22.0 (+3.5) | N/A | N/A |
| Qwen2.5-14B-Instruct (hybrid) | 21.0 (+4.0) | 21.0 (+2.5) | N/A | N/A |

instruction 2 derived from NNetNav, and with a maximum exploration budget of 30 steps. To extract environment dynamics from interaction logs, we summarize each state transition pair (observation, action taken, new observation) with prompt instruction 3, and concatenate all summarized state transition pairs to obtain the list of environmental dynamics $E = \{e\}_{i=1}^{M}$ for each website. Finally, we utilize the `o3` model to remove repetitive and trivial environment dynamic entries from the list via prompt instruction 4, resulting in $E_{\text{clean}}$. We provide all prompts we use in section A. With syntactic alignment, we reset the adaptation vector after each episode and we use a learning rate of 0.1 and training step of 1 for all experiments. During evaluation, we use the text-based accessibility tree as the observation type and run experiments using BrowserGym[3] and AgentLab[4] with a maximum step of 30.

**BFCLv3** BFCLv3 (Patil et al.) is a benchmark with various realistic APIs to evaluate the functions calling abilities of LLM. It contains 8 different API domains. We provide the number of APIs for each environment in table 8. We choose the multi-turn-base split to assess the agent's multi-turn function calling capabilities. This split is multi-turn and multi-step, meaning that instead of just handling a single tool call in a one-off request, models have to carry out sequences of function calls across multiple dialogue turns. BFCLv3 uses state-based evaluation (checking the system's resulting state) and response-based evaluation (checking whether the call sequences are valid and minimally sufficient) to assess correctness, therefore being stable and deterministic. For dynamics grounding, we synthesize 10 exploration tasks with `GPT-4o` conditioned on the function documentation for each domain via prompt instruction 7. Concretely, we instruct the LLM to think about *what functions it wants to explore in order to find out unpredictable outcomes or interesting behaviors, given the function documentation*. We use a `GPT-4.1` agent to execute the exploration task and we collect interaction logs for environment dynamics extraction. During exploration, we ask the exploration agent to decide if the steps taken so far are sufficient to answer the exploration task with prompt instruction 8. The environment dynamics are extracted using prompt instruction 9. We follow the same procedure as in WebArena to remove repetitive and trivial environment dynamics with prompt instruction 10. For syntactic alignment, we reset adaptation vector at each turn (when a new user query arrives).

**Tau-Bench** Tau-Bench (Yao et al., 2024) is a realistic function calling benchmark that simulates real-world service agents interacting with an LLM-simulated user. It comprises two domains: airline (50 tasks) and retail (115 tasks). The airline domain is generally more challenging than retail, as it features a much larger database, increasing the complexity of the tasks. Tau-bench's use of an LLM-simulated user agent introduces significant variance in evaluation results, as the simulated hu-

---

[3]https://github.com/ServiceNow/BrowserGym
[4]https://github.com/ServiceNow/AgentLab

Table 3: Success rates (%) on the WebArena benchmark across different websites and models. Both adaptation strategies improve performance over baselines. and dynamics grounding (DG) surpasses WMA on `GPT-4o mini`. DG improves success rate substantially on complex multi-site split.

| Model | Website | | | | | | |
| --- | --- | --- | --- | --- | --- | --- | --- |
| | Gitlab | Shop Admin | Shop | Reddit | Map | Multi | Avg |
| GPT-4.1 | 48.0% | 31.0% | 37.0% | 29.0% | 28.0% | 2.0% | 30.0% |
| GPT-4.1 (+DG) | 45.0% | 27.0% | 33.0% | 42.0% | 31.0% | 23.0% | 35.0% |
| GPT-4o mini | 9.0% | 12.0% | 18.0% | 8.0% | 12.0% | 12.0% | 12.0% |
| GPT-4o mini (+WMA) | – | – | – | – | – | – | 13.5% |
| GPT-4o mini (+DG) | 18.0% | 14.0% | 26.0% | 14.0% | 19.0% | 12.0% | 18.0% |
| Qwen2.5-14B-Instruct | 20.0% | 14.0% | 22.0% | 13.0% | 13.0% | 6.0% | 17.0% |
| Qwen2.5-14B-Instruct (+SA) | 16.0% | 14.0% | 24.0% | 25.0% | 16.0% | 6.0% | 18.0% |
| Qwen2.5-14B-Instruct (+DG) | 23.0% | 13.0% | 24.0% | 23.0% | 20.0% | 10.0% | 20.0% |
| Qwen2.5-14B-Instruct (Hybrid) | 26.0% | 13.0% | 24.0% | 31.0% | 19.0% | 4.0% | 21.0% |

man may drift from the original instruction or be influenced by the agent's responses over multiple conversational turns. To address this, we adopt the improved evaluation codebase from (Prabhakar et al., 2025), which substantially reduces reward variance and improves evaluation fidelity. Specifically, we (1) fix incorrect labels, (2) set the temperature of the LLM-simulated user agent to 0 for deterministic responses, and (3) employ a Best-of-N (N=5) sampling strategy combined with a self-critique mechanism, allowing the simulated user to better adhere to the task instruction and avoid being misled by the agent's outputs. This protocol has been shown to improve consistency in agent performance evaluation across multiple trials according to (Prabhakar et al., 2025). Tau-Bench consists solely of API-based tasks without observable state transitions, meaning there are no environment dynamics to extract. As a result, dynamics grounding, which relies on extracting and leveraging such dynamics, is not applicable to this benchmark. Therefore, we only evaluate the syntactic alignment method on Tau-Bench. To further reduce variance, we report results averaged over 5 random seeds in table 2.

## 4.2 Results and Analysis

**Both Adaptation Strategies Consistently Improve Performance**  As shown in table 2, both syntactic alignment (SA) and dynamics grounding (DG) improve performance across models and tasks. dynamics grounding yields substantial improvements for `GPT-4.1`, which is better able to leverage the provided dynamics due to its superior instruction-following abilities. It is also evident in table 2 that our `GPT-4o mini` surpasses WMA that uses a learned world model by a large margin. For `Qwen2.5-14B-Instruct`, both methods provide gains, highlighting the general applicability of test-time adaptation. The improvements of dynamics grounding are especially significant for models with stronger instruction-following abilities (e.g., GPT-4.1), likely because they can better leverage the provided dynamics context.

**Both Adaptation Strategies are computationally efficient**  The two proposed strategies are computationally efficient, though they have different profiles.  syntactic alignment is highly efficient for real-time use.  As shown in table 4, a single update step adds only a 3% latency overhead.  In contrast, dynamics grounding is a one-time, deployment-time investment designed to be far more lightweight than traditional world-modeling approaches like the WMA baseline. For example, while WMA requires synthesizing 870 tasks to collect trajectories for training a separate `Llama-3.1-8B` model, our method uses only 50 exploratory rollouts and requires no additional model training. This streamlined process avoids WMA's costly training phase and its significant inference overhead, where action simulation takes approximately 140 seconds per step. The total upfront cost for our dynamics grounding procedure on a single WebArena website is approximately 7M tokens, a cost that is amortized across all subsequent tasks. This investment pays significant dividends where environmental dynamics are unpredictable. For instance, on the challenging multi-site split, the success rate of our agent increases from 2% to 23%. This demonstrates the value of efficiently learning an environment's dynamics at test time to compensate for the limitations of an agent's pre-training.

Table 4: Relative latency gain (%) of PA on BFCLv3 multi-turn. LR fixed to $0.1$.

| Step | Latency Gain ↓ |
| --- | --- |
| 0 | 0% |
| 1 | 3.0% |
| 2 | 5.9% |
| 3 | 8.2% |
| 4 | 12.6% |
| 5 | 15.6% |

Table 5: Ablation on exploration policy and dynamics extractor backbones on WebArena. Here we use different backbones of exploration policy and dynamics extractor. We found for dynamics grounding—using the same LLM agent (itself) improves the same as using a stronger one.

| Task Model | Exploration Policy | Dynamics Extractor | SR |
|---|---|---|---|
| GPT-4o mini (baseline) | – | – | 12.0 |
| GPT-4o mini | GPT-4.1 | GPT-4.1 | 18.0 (+6.0) |
| GPT-4o mini (self) | GPT-4o mini | GPT-4o mini | 19.0 (+7.0) |
| Qwen2.5-14B-Instruct (baseline) | – | – | 17.0 |
| Qwen2.5-14B-Instruct | GPT-4.1 | GPT-4.1 | 20.0 (+3.0) |
| Qwen2.5-14B-Instruct (self) | Qwen2.5-14B-Instruct | Qwen2.5-14B-Instruct | 20.0 (+3.0) |

**Effectiveness Depends on Environment Complexity and Method Synergy** The smaller gains for DG on simpler websites like "Shopping" suggest that when an environment's dynamics align with common sense priors (e.g., clicking 'search' shows a search bar), the explicit dynamics provide less new information, and the overhead of a longer context may slightly hinder performance. In these simple settings, longer input on dynamics may negatively impact the model's effectiveness, as prior work has shown that excessively long contexts can hurt performance (Beltagy et al., 2020; Liu et al., 2024). We further analyze this phenomenon and provide exemplar environment dynamics in section A.5. Furthermore, our test of a naive hybrid approach showed mixed results, underperforming the dynamics grounding method alone on BFCLv3 (21.0% vs 22.0%). This suggests a simple combination is insufficient and may create interference between the two adaptation signals. A key challenge for future work is developing a more principled integration of semantic guidance and distributional alignment.

## 4.3 ABLATIONS

**Dynamics grounding enables self-improvement** We ablate the choice of backbone LLMs for exploration and dynamics extraction in our dynamics grounding method (table 5). Results show that using the same model for both components (self-improvement) performs as well as using a stronger model, indicating that our approach is robust to the choice of exploration policy and dynamics extractor.

**Effect of filtering environmental dynamics** We conduct an ablation experiment on BFCLv3 and find that filtering improves success rate across exploration budgets. Specifically, with 10 episodes, success rate rises from 61.0 to 64.0 (+3.0).

**Syntactic alignment is robust to different hyperparameters** As shown in table 6, training with different learning rates generally improves performance across a range of reasonable hyperparameter values. However, using extreme hyperparameters can degrade performance. For example, in the case of the 7B model, applying a very high learning rate of 1.0 or training for 5 steps leads to reduced success rates, highlighting the importance of careful hyperparameter selection for optimal adaptation.

**Larger models benefit from a larger learning rate** When testing with a very large learning rate (i.e., learning rate of 1.0), we find that larger models (i.e., 14B and 32B models) benefit more with a larger learning rate. We attribute this to the fact that the dimension of the adaptation vector for 14B and 32B models is the same (i.e., 5120), and is much larger compared to the 7B model's (i.e., 3584).

## 5 LIMITATIONS AND FUTURE WORK

Our study is primarily focused on the Qwen2.5 model family for syntactic alignment; future work should validate these findings across a broader range of open-source architectures. Our dynamics grounding method currently operates in environments with explicit state transitions, where testbeds such as user-oriented conversational tasks are not explored. Additionally, our online syntactic alignment method does not currently normalize updates by the hidden dimension size, which could improve robustness to hyperparameters.

The most promising direction for future work is the development of a principled method for integrating our two adaptation strategies as our analysis suggests that simply combining them is suboptimal. A more advanced approach could involve a meta-controller that assesses an environment's complexity to dynamically decide whether to rely on efficient online adaptation for simple tasks or to deploy the more costly dynamics grounding for complex, unfamiliar environments.

## 6 CONCLUSIONS

In this work, we presented a comparative study of two distinct strategies for adapting LLM agents to novel, complex environments. We identified two primary failure modes—syntactic and semantic mismatches—and investigated targeted solutions for each. The first, syntactic alignment (SA), is an efficient technique for aligning an agent's output distribution to the local format of an environment. The second, dynamics grounding (DG), is an effective and efficient method for providing the agent with a causal world model through proactive dynamics extraction.

Our evaluations on diverse agentic benchmarks demonstrate that both strategies significantly improve performance. Online adaptation provides consistent, low-cost gains, while dynamics grounding is critical for success in complex settings where an agent's pre-trained knowledge fails. Our findings highlight that a multi-faceted approach to adaptation is necessary for robust agent generalization and point toward exciting future directions in creating more intelligent and integrated adaptation systems.

## ACKNOWLEDGMENTS

We thank Salesforce AI Research for supporting this work. We also thank the reviewers and the area chairs for their time and thoughtful feedback.

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

# A APPENDIX

## A.1 EXPERIMENT DETAILS

**WebArena** We follow the BrowserGym guide[5] to setup WebArena enviornment. We use AWS EC2 instance with a `t3a.xlarge` instance. For all exploration episodes of dynamics grounding, we use a temperatue of 1.0. We use a temperature of 0.0 for all task evaluation experiments.

**BFCLv3** For BFCLv3, we follow the official setup[6] and consistently using a temperature of 0.

---

[5] `https://github.com/ServiceNow/BrowserGym/blob/main/browsergym/webarena/README.md`

[6] `https://github.com/ShishirPatil/gorilla/tree/main/berkeley-function-call-leaderboard`

**Tau-bench** We adopt the codebase form (Prabhakar et al., 2025) and use Best-of-N (N=5) strategy for LLM-simulated user agent, so each trajectory will be scored by an LLM judge and the reponse that best adheres with user intent will be select, therefore drastically mitigating variance in average reward.

## A.2 PROMPTS USED IN WEBARENA EXPERIMENTS

Listing 1: prompt used to synthesize personas for WebArena

```
You are an expert at creating realistic user personas given a
    website. You are given the name of a website and some example
    outputs for the website, generate a diverse list of ${
    n_personas} personas, each with a unique "persona" name and a
    1-2 sentence "description" of their typical behavior,
    interests, or motivations on that site.

Format your output as a JSON array, where each element is an
    object with two fields:
- "persona": a short, descriptive name for the persona (e.g., "
    Casual Browser", "Tech Enthusiast")
- "description": a brief description of how this persona typically
     uses the website

Now, generate a list of personas in the specified JSON format. You
     are given the name of the website and some example outputs
    for the website.
Website: ${website}
Example outputs: ${examples}
```

Listing 2: prompt used for persona-driven exploration in WebArena

```
You are an autonomous intelligent agent tasked with navigating a
    web browser. Your objective is to simulate a task that a
    person might perform, by interacting with the browser through
    the use of specific actions.

Here's the information you'll have:

The current web page's accessibility tree: This is a simplified
    representation of the webpage, providing key information.
The current web page's URL: This is the page you're currently
    navigating.
The open tabs: These are the tabs you have open.
The previous action: This is the action you just performed. It may
     be helpful to track your progress.
Trajectory: This is a sequence of natural language descriptions of
     the agent's interaction with the web-browser.
Person Description: The description of a specific kind of person
    whose task you are supposed to simulate.
Environment dynamics: Descriptions of how states transition.

The actions you can perform fall into several categories:

Page Operation Actions:
'click [id]': This action clicks on an element with a specific id
    on the webpage.
'type [id] [content] [press_enter_after=0|1]': Use this to type
    the content into the field with id. By default, the "Enter"
    key is pressed after typing unless press_enter_after is set to
     0.
```

```
'hover [id]': Hover over an element with id.
'press [key_comb]': Simulates the pressing of a key combination
    on the keyboard (e.g., Ctrl+v).
'scroll [direction=down|up]': Scroll the page up or down.

Tab Management Actions:
'new_tab': Open a new, empty browser tab.
'tab_focus [tab_index]': Switch the browser's focus to a specific
    tab using its index.
'close_tab': Close the currently active tab.

URL Navigation Actions:
'goto [url]': Navigate to a specific URL.
'go_back': Navigate to the previously viewed page.
'go_forward': Navigate to the next page (if a previous 'go_back'
    action was performed).

Completion Action:
'stop ["done"]': Issue this action when you are done.

Homepage:
If you want to visit other websites, check out the homepage at
    http://homepage.com. It has a list of websites you can visit.

To be successful, it is very important to follow the following
    rules:
1. You should only issue an action that is valid given the current
    observation
2. You should only issue one action at a time.
3. You should follow the examples to reason step by step and then
    issue the next action.
4. Generate the action in the correct format. Start with a "In
    summary, the next action I will perform is" phrase, followed
    by action inside '''''. For example, "In summary, the next
    action I will perform is '''click [1234]'''".
5. To generate an interesting task, make sure you issue atleast 4
    actions before stopping. More interesting tasks typically
    involve more interactions with the browser.
6. You can issue atmost 40 actions before stopping, but feel free
    to output the stop action early if you want to stop exploring.
     Don't generate anything after stop.
Here are the list of environmental dynamics of this environment:

OBSERVATION:
[1744] link 'HP CB782A#ABA 640 Inkjet Fax Machine (Renewed)'
                [1749] StaticText '$279.49'
                [1757] button 'Add to Cart'
                [1760] button 'Add to Wish List'
                [1761] button 'Add to Compare'
URL: http://onestopmarket.com/office-products/office-electronics.
    html
ENVIRONMENTAL DYNAMICS:
[{'initial_state': 'Homepage of onestopmarket.com, with the button
     Add to Cart visible', 'action': 'click [1757] where the
    button is purchase'}]
PREVIOUS ACTION: None
Let's think step-by-step. This page lists the information of HP
    Inkjet Fax Machine, which is the product identified in the
    objective. Its price is $279.49. I have checked the
```

```
        environmental dynamics and see that stopping with this answer
        from this state has not been done before, so this action will
        contribute to exploration diversity. I think I have achieved
        the objective. In summary, the next action I will perform is
        '''stop [$279.49]'''
OBSERVATION:
[164] textbox 'Search' focused: True required: False
[171] button 'Go'
[174] link 'Find directions between two points'
[212] heading 'Search Results'
[216] button 'Close'
URL: http://openstreetmap.org
ENVIRONMENTAL DYNAMICS:
[{'initial_state': 'OpenStreetMap homepage with search box and Go
        button visible', 'action': 'click [171] to submit a search'},
        {'initial_state': 'OpenStreetMap homepage with search box
        focused', 'action': 'type [164] [parks near NY] [1]'}]
PREVIOUS ACTION: None
Let's think step-by-step. This page has a search box whose ID is
        [164]. According to the nominatim rule of openstreetmap, I can
         search for restaurants near a location by typing "restaurants
         near" followed by the location. I can submit my typing by
        pressing Enter afterwards. Before proceeding, I will check the
         environmental dynamics to see if a similar action has already
         been performed from this state. I see that there is no
        previous action where, starting from the current state,
        someone searched for "restaurants near CMU" in the search box.
         This means my action will contribute to exploration diversity
        . In summary, the next action I will perform is '''type [164]
        [restaurants near CMU] [1]'''
OBSERVATION:
${OBSERVATION}
Person Description:
${PERSON DESCRIPTION}
URL: ${URL}
PREVIOUS ACTION: ${PREVIOUS ACTION}
ENVIRONMENTAL DYNAMICS:
${ENVIRONMENTAL DYNAMICS}
```

Listing 3: prompt used to extract environment dynamics from observations

```
You are given the output of an action taken by an autonomous
        intelligent agent navigating a web browser. Your objective is
        to produce a description of the changes made to the state of
        the browser. Specifically, you need to produce the following
        elements:
1) Initial state: provide a concise description of the browser's
        state before the action was taken (for example: "GitHub
        homepage" or "GitHub repository page").
2) Environmental dynamics: clearly describe the specific changes
        that occurred in the browser's state as a result of the action
         (for example: "The search bar expanded and is able to consume
         input").

Here's the information you'll have:
- **Initial state of the browser as an accessibility tree:**
  This is a simplified representation of the webpage, providing
        key information.
- **Final state of the browser:**
```

```
    This is the accessibility tree representation after the agent
        took the action.
- **The action taken by the web agent:**
  The agent can take actions that fall under the following
      categories (with descriptions):
**Page Operation Actions:**
- 'click [id]': Clicks on an element with a specific id on the
    webpage.
- 'type [id] [content] [press_enter_after=0|1]': Types the content
      into the field with id. By default, the "Enter" key is
    pressed after typing unless 'press_enter_after' is set to 0.
- 'hover [id]': Hovers over an element with id.
- 'press [key_comb]': Simulates pressing a key combination on the
    keyboard (e.g., Ctrl+v).
- 'scroll [direction=down|up]': Scrolls the page up or down.

**Tab Management Actions:**
- 'new_tab': Opens a new, empty browser tab.
- 'tab_focus [tab_index]': Switches the browser's focus to a
    specific tab using its index.
- 'close_tab': Closes the currently active tab.

**URL Navigation Actions:**
- 'goto [url]': Navigates to a specific URL.
- 'go_back': Navigates to the previously viewed page.
- 'go_forward': Navigates to the next page (if a previous 'go_back
    ' action was performed).

**Completion Action:**
- 'stop [answer]': Issue this action when you believe the task is
    complete. If the objective is to find a text-based answer,
    provide the answer in the bracket. If you believe the task is
    impossible to complete, provide the answer as "N/A" in the
    bracket.

**To be successful, it is very important to follow these rules:**
1. Explicitly think about the various features on the website and
    how the interaction with the website changed these features.
2. Provide the description of changes in one or two sentences.
3. If there is no change, your description of the changes in state
      should be "no change".
4. Strictly follow the JSON format in your response: {"
    initial_state": <description of initial state>, "
    environmental_dynamics": <description of environmental change
    >}

Here are some output examples for some random tasks:
1. {"initial_state": "Google homepage with search bar visible", "
    environmental_dynamics": "A modal search dialog opened up,
    bringing up a focused search input with suggestions, allowing
    the user to search or jump to various GitHub sections."}
2. {"initial_state": "Wikipedia article page", "
    environmental_dynamics": "The page scrolled down, revealing
    the 'History' section."}
3. {"initial_state": "Amazon product page", "
    environmental_dynamics": "A confirmation message appeared
    indicating the item was added to the cart."}
```

Listing 4: prompt used to remove repeatitive or trivial environment dynamics in WebArena

```
You are given a list of environmental dynamics of an environment
    collected by an autonomous web browsing agent. You are tasked
    to remove repetitive and trivial environmental dynamics. Each
    of the point consists of the following:
1) initial_state: provide a concise description of the browser's
    state before the action was taken (for example: "GitHub
    homepage" or "GitHub repository page").
2) action: action taken to make the environmental change
2) environmental_dynamics: clearly describe the specific changes
    that occurred in the browser's state as a result of the action
     (for example: "The search bar expanded and is able to consume
     input").

To be successful, it is important to follow these rules:
1. entries with trivial enviornmental dynamics, such as "when I
    scroll down, the page will reveal more content" should be
    removed. Trivial environmental dynamics are those that do not
    reflect a meaningful or unique change in the environment, such
     as simple scrolling, pagination, or toggling the visibility
    of already available content without introducing new
    information or interface elements.

2. Remove entries where the environmental_dynamics only describe
    the expansion or collapse of text (e.g., "The full abstract
    expanded and is now visible"), unless the expansion reveals
    new interface elements or options that were not previously
    accessible.

3. Remove entries where the environmental_dynamics only describe
    navigation to a different section of the same page (e.g., "The
     page scrolled to the 'References' section"), unless this
    navigation results in a substantial change in the available
    interface or content.

4. Keep entries where the environmental_dynamics describe:
    - Unexpected or spurious state change of executing an action at
        the initial state.
    - Keep entries where the environmental_dynamics indicate that
        performing an action did not result in any change to the
        browser's state, even though a change would normally be
        expected (for example, clicking a button that should open a
        menu but nothing happens).
    - The appearance of new controls, filters, or options in the
        interface.
    - Navigation to a new page or a substantially different view (e
        .g., from search results to a detailed article page).
    - The addition or removal of search fields, filters, or other
        interactive elements.
    - Any change that enables new actions or reveals new types of
        information not previously accessible.

5. When in doubt, prefer to remove entries that are repetitive (e.
    g., multiple entries describing the same type of abstract
    expansion) or that do not add new information about the
    environment's capabilities or state.
```

```
Your output should be a cleaned list of environmental dynamics,
    with only the non-trivial, non-repetitive, and meaningful
    changes retained. Please output in the original JSON format.
```

Listing 5: prompt used for evaluation in WebArena

```
You are an autonomous intelligent agent tasked with navigating a
    web browser. You will be given web-based tasks. These tasks
    will be accomplished through the use of specific actions you
    can issue.

Here's the information you'll have:
The user's objective: This is the task you're trying to complete.
The current web page's accessibility tree: This is a simplified
    representation of the webpage, providing key information.
The current web page's URL: This is the page you're currently
    navigating.
The open tabs: These are the tabs you have open.
The previous actions: These are all the action you have performed.
     It may be helpful to track your progress.

The actions you can perform fall into several categories:

Page Operation Actions:
'click [id]': This action clicks on an element with a specific id
    on the webpage.
'type [id] [content] [press_enter_after=0|1]': Use this to type
    the content into the field with id. By default, the "Enter"
    key is pressed after typing unless press_enter_after is set to
     0.
'hover [id]': Hover over an element with id.
'press [key_comb]': Simulates the pressing of a key combination
    on the keyboard (e.g., Ctrl+v).
'scroll [down|up]': Scroll the page up or down.

Tab Management Actions:
'new_tab': Open a new, empty browser tab.
'tab_focus [tab_index]': Switch the browser's focus to a specific
    tab using its index.
'close_tab': Close the currently active tab.

URL Navigation Actions:
'goto [url]': Navigate to a specific URL.
'go_back': Navigate to the previously viewed page.
'go_forward': Navigate to the next page (if a previous 'go_back'
    action was performed).

Completion Action:
'stop [answer]': Issue this action when you believe the task is
    complete. If the objective is to find a text-based answer,
    provide the answer in the bracket. If you believe the task is
    impossible to complete, provide the answer as "N/A" in the
    bracket.

Homepage:
If you want to visit other websites, check out the homepage at
    http://homepage.com. It has a list of websites you can visit.
```

```
To be successful, it is very important to follow the following
    rules:
1. You should only issue an action that is valid given the current
    observation
2. You should only issue one action at a time.
3. You should follow the examples to reason step by step and then
    issue the next action.
4. You are strictly forbidden from issuing a goto action to a URL
    that is not on the homepage.
5. Generate the action in the correct format. Start by reasoning
    about the current situation. End with "In summary, the next
    action I will perform is" phrase, followed by action inside
    '''''. For example, "Let's think step-by-step. Given the
    current state, I need to click on the like button which has id
     1234. In summary, the next action I will perform is '''click
    [1234]'''".
6. Issue stop action when you think you have achieved the
    objective. Don't generate anything after stop.

Here are some example outputs for some random tasks:
1. Let's think step-by-step. This page list the information of HP
    Inkjet Fax Machine, which is the product identified in the
    objective. Its price is $279.49. I think I have achieved the
    objective. I will issue the stop action with the answer. In
    summary, the next action I will perform is '''stop [$279
    .49]'''
2. Let's think step-by-step. This page has a search box whose ID
    is [164]. According to the nominatim rule of openstreetmap, I
    can search for the restaurants near a location by "restaurants
     near". I can submit my typing by pressing the Enter
    afterwards. In summary, the next action I will perform is '''
    type [164] [restaurants near CMU] [1]'''
```

Listing 6: prompt used for evaluation on WebArena with environment dynamics

```
You are an autonomous intelligent agent tasked with navigating a
    web browser. You will be given web-based tasks. These tasks
    will be accomplished through the use of specific actions you
    can issue.

Here's the information you'll have:
The user's objective: This is the task you're trying to complete.
The current web page's accessibility tree: This is a simplified
    representation of the webpage, providing key information.
The current web page's URL: This is the page you're currently
    navigating.
The open tabs: These are the tabs you have open.
The previous actions: These are all the action you have performed.
     It may be helpful to track your progress.
The environmental dynamics: You will also be given the
    environmental dynamics of the environment. You can use the
    environmental dynamics to predict the next action. Each
    environmental dynamics entry describes a state change in the
    environment and contains the following fields:
- "initial_state": The state of the environment before the action
    is taken.
- "action": The action performed by the agent.
```

```
- "environmental_dynamics": A description of how the environment
    changes as a result of the action.

The actions you can perform fall into several categories:

Page Operation Actions:
'click [id]': This action clicks on an element with a specific id
    on the webpage.
'type [id] [content] [press_enter_after=0|1]': Use this to type
    the content into the field with id. By default, the "Enter"
    key is pressed after typing unless press_enter_after is set to
     0.
'hover [id]': Hover over an element with id.
'press [key_comb]': Simulates the pressing of a key combination
    on the keyboard (e.g., Ctrl+v).
'scroll [down|up]': Scroll the page up or down.

Tab Management Actions:
'new_tab': Open a new, empty browser tab.
'tab_focus [tab_index]': Switch the browser's focus to a specific
    tab using its index.
'close_tab': Close the currently active tab.

URL Navigation Actions:
'goto [url]': Navigate to a specific URL.
'go_back': Navigate to the previously viewed page.
'go_forward': Navigate to the next page (if a previous 'go_back'
    action was performed).

Completion Action:
'stop [answer]': Issue this action when you believe the task is
    complete. If the objective is to find a text-based answer,
    provide the answer in the bracket. If you believe the task is
    impossible to complete, provide the answer as "N/A" in the
    bracket.

Homepage:
If you want to visit other websites, check out the homepage at
    http://homepage.com. It has a list of websites you can visit.

To be successful, it is very important to follow the following
     rules:
1. You should only issue an action that is valid given the current
     observation
2. You should only issue one action at a time.
3. You should follow the examples to reason step by step and then
    issue the next action.
4. You are strictly forbidden from issuing a goto action to a URL
    that is not on the homepage.
5. Generate the action in the correct format. Start by reasoning
    about the current situation. End with "In summary, the next
    action I will perform is" phrase, followed by action inside
    '''''. For example, "Let's think step-by-step. Given the
    current state, I need to click on the like button which has id
     1234. In summary, the next action I will perform is '''click
    [1234]'''".
6. When selecting actions, consider the environmental dynamics -
    what state changes will occur based on your knowledge of how
    the website behaves. Use this to avoid actions that would lead
```

```
        to undesired states or to strategically choose actions that
        lead to desired outcomes.
7.  Issue stop action when you think you have achieved the
        objective. Don't generate anything after stop.

Here are some example outputs for some random tasks:
1.  Let's think step-by-step. This page lists the information of HP
        Inkjet Fax Machine, which is the product identified in the
        objective. Its price is $279.49. By considering the
        environmental dynamics, I can confirm that no further actions
        are needed to change the state, and I have achieved the
        objective. I will issue the stop action with the answer. In
        summary, the next action I will perform is '''stop [$279
        .49]'''
2.  Let's think step-by-step. This page has a search box whose ID
        is [164]. According to the nominatim rule of openstreetmap, I
        can search for the restaurants near a location by "restaurants
        near". By considering the environmental dynamics, I know that
        typing this query and pressing Enter will trigger the search
        and update the results accordingly. Therefore, I can submit my
        typing by pressing the Enter afterwards. In summary, the next
        action I will perform is '''type [164] [restaurants near CMU]
        [1]'''
```

## A.3 PROMPTS USED IN BFCLV3 EXPERIMENTS

Listing 7: prompt used to synthesize exploration tasks on BFCLv3

```
You are an expert at creating exploration strategies that will
    help guide an LLM to explore and understand a new function
    calling environment. Your task is to synthesize some
    exploration goals based on a list of functions and a specific
    environment.

## Task
Given a list of functions and an environment, create ${N} distinct
    goals that will explore diffferent aspects of the environment
    . Each goal should represent different exploration strategies
    and approaches to discovering functionality.

## Input Format
- **Environment**: The computing environment or system (e.g., "
    file system", "vehicle control", "web browser")
- **Functions**: A list of available functions/commands alongside
    with detailed descriptions (e.g., ["cd()", "ls()", "pwd()", "
    mkdir()", "rm()"])

## Output Format
Output a JSON list of goals. Each goal should be a single sentence
    that captures the exploration goal and the strategy to
    achieve it.

Format: ["goal description", "goal description", ...]

## Example Output
**Environment**: File System
**Functions**:
{"name": "touch", "description": "This tool belongs to the Gorilla
    file system. It is a simple file system that allows users to
    perform basic file operations such as navigating directories,
    creating files and directories, reading and writing to files,
    etc. Tool description: Create a new file of any extension in
    the current directory.", "parameters": {"type": "dict", "
    properties": {"file_name": {"type": "string", "description": "
    The name of the new file in the current directory. file_name
    is local to the current directory and does not allow path."}},
     "required": ["file_name"]}, "response": {"type": "dict", "
    properties": {}}}
{"name": "wc", "description": "This tool belongs to the Gorilla
    file system. It is a simple file system that allows users to
    perform basic file operations such as navigating directories,
    creating files and directories, reading and writing to files,
    etc. Tool description: Count the number of lines, words, and
    characters in a file of any extension from current directory
    .", "parameters": {"type": "dict", "properties": {"file_name":
     {"type": "string", "description": "Name of the file of
    current directory to perform wc operation on."}, "mode": {"
    type": "string", "description": "Mode of operation ('l' for
    lines, 'w' for words, 'c' for characters). ", "default": "l
    "}}, "required": ["file_name"]}, "response": {"type": "dict",
    "properties": {"count": {"type": "integer", "description": "
    The count of the number of lines, words, or characters in the
    file."}, "type": {"type": "string", "description": "The type
```

```
        of unit we are counting. [Enum]: [\"lines\", \"words\", \"
        characters\"]"}}}}
{"name": "ls", "description": "This tool belongs to the Gorilla
        file system. It is a simple file system that allows users to
        perform basic file operations such as navigating directories,
        creating files and directories, reading and writing to files,
        etc. Tool description: List the contents of the current
        directory.", "parameters": {"type": "dict", "properties": {"a
        ": {"type": "boolean", "description": "Show hidden files and
        directories. Defaults to False. ", "default": false}}, "
        required": []}, "response": {"type": "dict", "properties": {"
        current_directory_content": {"type": "array", "description": "
        A list of the contents of the specified directory.", "items":
        {"type": "string"}}}}}
{"name": "rm", "description": "This tool belongs to the Gorilla
        file system. It is a simple file system that allows users to
        perform basic file operations such as navigating directories,
        creating files and directories, reading and writing to files,
        etc. Tool description: Remove a file or directory.", "
        parameters": {"type": "dict", "properties": {"file_name": {"
        type": "string", "description": "The name of the file or
        directory to remove. "}}, "required": ["file_name"]}, "
        response": {"type": "dict", "properties": {"result": {"type":
        "string", "description": "The result of the remove operation
        ."}}}}

Generated goals:
[
   "Explore simple ways to create a file with touch(), count words
        of the file with wc(), and remove the file with rm().",
   "Understand the "file_name" argument of the touch() function to
        see if a file can be created using a path instead of just a
        file name."
]

## Guidelines
- **Be function-specific**: Each goal should explicitly mention
    which specific functions/tools will be used and in what
    sequence
- **Incorporate function descriptions into goal formulation**: You
    should examine the given function descriptions (i.e., input
    or output arguments) to propose goals to solve confusing or
    ambiguous function descriptions or environmental dynamics.
- **Ensure clarity and achievability**: Each goal should be a
    single, clear action that can be completed with the available
    functions

## Your Task
Now synthesize goal descriptions for the environment and functions
    provided in the format of a JSON list. Please do not leave
    any comment.
```

Listing 8: user prompt used in BFCLv3 exploration

```
Do you think you have well-explored all functions that you are
    interested in, particularly those with ambiguous input formats
    /structures or potentially unexpected outputs?
```

```
Consider stopping (###STOP) if:
- You have thoroughly tested functions with unclear parameter
    requirements
- You have explored functions that might produce unexpected or
    variable output formats
- You have sufficient understanding of how to handle edge cases
    and errors

Continue exploring (###CONTINUE) if:
- There are still functions with ambiguous input specifications
    that need clarification
- You haven't fully tested functions that could have unpredictable
    outputs
- You need more exploration to understand parameter constraints
    and error handling

Please output ###STOP if you believe you have comprehensive
    coverage, or ###CONTINUE if more exploration is needed.
```

Listing 9: prompt used to extract environment dynamics from interaction logs of BFCLv3

```
You are an expert data analyst specializing in function calling
    environments. Your task is to analyze interaction logs between
     a function calling bot and an environment to extract
    environmental dynamics based on a given exploration goal.

## Task
Given an exploration goal and an interaction log, extract
    environmental dynamics that describe how the environment
    changes in response to bot actions. Each environmental dynamic
     should capture the cause-and-effect relationship between
    actions and environmental changes.

## Environmental Dynamic Structure
Each environmental dynamic must include these three components:

1. **initial_state**: The relevant state of the environment before
     the action
   - Include context like current directory, variable values, file
        states, or any pertinent environmental conditions
   - Be specific about what matters for understanding the action's
        impact

2. **action_taken**: The exact function call or action performed
    by the bot
   - Include the function name and all parameters with their
        values
   - Use the exact syntax from the interaction log

3. **environmental_dynamics**: How the environment changes as a
    result of the action
   - Describe what was created, modified, deleted, or revealed
   - Include any state transitions, data changes, or new
        information made available
   - Be specific about the observable effects

## Guidelines
- **Output format**: Generate your response as a JSON array of
    objects
```

```
– **Completeness**: Extract dynamics for every significant action
    in the log
– **Accuracy**: Base descriptions strictly on what's observable in
    the interaction log
– **Clarity**: Use precise, unambiguous language for each
    component
– **Focus**: Only include dynamics that show meaningful
    environmental changes

## Example Input
Goal: Investigate how to create a new file with touch() and
    confirm its existence using ls().

Interaction Logs:
Step 1:
Assistant: [ls()]
Environment: {"current_directory_content": []}
Step 2:
Assistant: [mkdir(dir_name="testdir")]
Environment: None
Step 3:
Assistant: [cd(folder="testdir")]
Environment: {"current_working_directory": "testdir"}
Step 4:
Assistant: [pwd()]
Environment: {"current_working_directory": "///testdir"}

## Example Output
[
  {
    "initial_state": "The current directory contains no files.",
    "action_taken": "touch(file_name=\"testfile.txt\")",
    "environmental_dynamics": "A new file named \"testfile.txt\"
        is created in the current directory."
  },
  {
    "initial_state": "The current directory contains the file \"
        testfile.txt\".",
    "action_taken": "ls()",
    "environmental_dynamics": "The environment reveals the
        contents of the current directory, showing that \"testfile
        .txt\" exists."
  }
]
```

Listing 10: prompt used to filter environment dynamics of BFCLv3

```
You are an expert data analyst specializing in function calling
    environments. Your task is to filter environmental dynamics to
     remove insignificant or environment−specific details while
    preserving meaningful behavioral patterns.

## Task
Given a JSON array of environmental dynamics, identify and remove
    dynamics that are:
1. **Very environment−specific**: Contain specific data, names, or
     values that are unique to a particular environment instance/
     setup
```

2. **Insignificant**: Describe trivial or non-meaningful changes that don't contribute to understanding function behavior (e.g., ls() in a file system)
3. **Redundant**: Duplicate similar patterns already captured in other dynamics

## Filtering Criteria

### Remove dynamics that contain:
- **Specific data values**: Names, IDs, file contents, user lists, or any concrete data that varies between environments
- **Environment-specific paths**: Absolute paths, specific directory names, or location-dependent information
- **Instance-specific identifiers**: User names, session IDs, timestamps, or other unique identifiers
- **Trivial state changes**: Minor formatting changes, temporary states, or cosmetic updates
- **Redundant information**: Multiple dynamics describing the same behavioral pattern with different specific values

### Keep dynamics that describe:
- **General behavioral patterns**: How functions typically behave across different environments
- **Error conditions**: Standard error messages and failure modes
- **State transitions**: General state changes (logged in/out, file created/deleted, etc.)
- **Function capabilities**: What functions can do, not what specific data they return
- **Function contains meaningful arguments**: Keep dynamics where the function call includes arguments that affect behavior or state, as these illustrate how input parameters influence outcomes.

## Guidelines
- **Output format**: Return a JSON array containing only the filtered dynamics
- **Preserve structure**: Keep the same JSON structure for remaining dynamics
- **Generalize descriptions**: Replace specific values with general descriptions where possible
- **Maintain accuracy**: Don't change the core meaning of the dynamics, only remove specificity
- **Focus on patterns**: Prioritize dynamics that show reusable behavioral patterns

## Examples

### Remove (Very environment-specific that it contains data entries specific to a environment instance):
```json
{
    "initial_state": "No user is logged into the system, and a list of users needs to be identified.",
    "action_taken": "list_users()",
    "environmental_dynamics": "Provides the list of available users: Alice, Bob, Catherine, and Daniel."
}
```

```
### Keep (General pattern and error message):
```json
{
    "initial_state": "No user is currently logged in.",
    "action_taken": "search_messages(keyword=\"test\")",
    "environmental_dynamics": "An error message is returned
        indicating that no user is currently logged in."
}
```

### Remove (Trivial transition):
```json
{
    "initial_state": "In a directory with existing files.",
    "action_taken": "ls()",
    "environmental_dynamics": "Returns a list of files and
        directories in the current location."
}
```

## Instructions
1. Analyze each environmental dynamic in the input JSON array
2. Apply the filtering criteria to determine if it should be kept
   or removed
3. For dynamics that are kept, generalize any remaining specific
   details
4. Return the filtered JSON array with only the significant,
   generalizable dynamics
```

## A.4    ADDITIONAL RESULTS

Table 6: Success rate (SR) of syntactic alignment across different Qwen2.5 instruct model sizes (7B, 14B, 32B) and training hyperparameters. Results are shown for varying numbers of training iterations and two learning rates (LR=0.1 and LR=1.0). Syntactic alignment adaptation generally improves SR over the baseline, with larger models and moderate learning rates yielding higher gains.

| Model | Number of Training Iterations | SR (LR=0.1) | SR (LR=1.0) |
|---|---|---|---|
| Qwen2.5-7B-Instruct | Baseline (0) | 9.00 | - |
| Qwen2.5-7B-Instruct | 1 | 9.50 | 9.50 |
| Qwen2.5-7B-Instruct | 2 | 9.50 | 8.50 |
| Qwen2.5-7B-Instruct | 3 | 9.50 | 9.00 |
| Qwen2.5-7B-Instruct | 4 | 10.00 | 8.50 |
| Qwen2.5-7B-Instruct | 5 | 10.00 | 8.50 |
| Qwen2.5-14B-Instruct | Baseline (0) | 18.5 | - |
| Qwen2.5-14B-Instruct | 1 | 20.00 | 20.00 |
| Qwen2.5-14B-Instruct | 2 | 20.00 | 20.00 |
| Qwen2.5-14B-Instruct | 3 | 20.00 | 21.00 |
| Qwen2.5-14B-Instruct | 4 | 19.50 | 20.00 |
| Qwen2.5-14B-Instruct | 5 | 19.00 | 19.00 |
| Qwen2.5-32B-Instruct | Baseline (0) | 26.00 | - |
| Qwen2.5-32B-Instruct | 1 | 25.50 | 26.50 |
| Qwen2.5-32B-Instruct | 2 | 26.50 | 26.50 |
| Qwen2.5-32B-Instruct | 3 | 26.50 | 27.00 |
| Qwen2.5-32B-Instruct | 4 | 26.50 | 27.00 |
| Qwen2.5-32B-Instruct | 5 | 26.00 | 26.50 |

Table 7: Number of environment dynamics per website before and after filtering with `GPT-4.1` as the exploration policy and dynamics extractor. Most environment dynamics are filtered out.

| Website | Number of Env Dynamics | Number of Env Dynamics (after filtering) |
|---|---|---|
| Shopping | 334 | 52 |
| Shopping Admin | 334 | 50 |
| GitLab | 334 | 53 |
| Map | 334 | 18 |
| Reddit | 334 | 23 |

Table 8: Number of functions available in each BFCLv3 environment.

| Environment | Num of Functions |
|---|---|
| VehicleControlAPI | 22 |
| GorillaFileSystem | 18 |
| TravelAPI | 17 |
| MathAPI | 17 |
| TwitterAPI | 14 |
| MessageAPI | 10 |
| TicketAPI | 9 |
| TradingBot | 22 |

## A.5 ENVIRONMENT DYNAMICS ANALYSIS

Here we provide some concrete examples of environment dynamics that are removed by our filtering mechanism. Exploration policy and dynamics extractor is `GPT-4.1`.

- **No Change Dynamics**

```
{
  "initial_state": "Magento Admin product edit page, with the '
      Country of Manufacture' dropdown expanded and no country
      selected.",
  "action": "click [1405] where [1405] is United States",
  "environmental_dynamics": "no change"
}
```

- **Basic Dropdown/Combobox Expansion**

```
{
  "initial_state": "Magento Admin product edit page for 'Sprite
      Stasis Ball 65 cm', with the 'Country of Manufacture'
      combobox (notice-WPNSU51) collapsed and not focused.",
  "action": "click [1168] where [1168] is notice-WPNSU51",
  "environmental_dynamics": "The 'Country of Manufacture' combobox
       became focused and expanded, showing its list of country
      options."
}
```

- **Basic Dialog Open/Close**

```
{
  "initial_state": "Magento Admin product edit page for 'Sprite
      Stasis Ball 65 cm', with the 'Add Attribute' button visible
      but no attribute selection dialog open.",
  "action": "click [720] where [720] is Add Attribute",
  "environmental_dynamics": "Clicking the 'Add Attribute' button
      opened the 'Add Attribute' dialog, displaying available
      product attributes for selection and actions such as 'Create
      New Attribute'."
}
```

- **Standard Form Field Visibility**

```
{
  "initial_state": "Magento Admin product detail page for 'Sprite
      Stasis Ball 65 cm' with all product attribute forms visible,
      including a collapsed 'Country of Manufacture' dropdown (
      combobox).",
  "action": "click [1267] where [1267] is notice-BKW2147",
  "environmental_dynamics": "The 'Country of Manufacture' combobox
       was clicked, causing attribute fields for 'Country of
      Manufacture' and all subsequent grouped attributes to become
      visible."
}
```

- **Basic Button Focus**

```
{
  "initial_state": "Magento Admin product edit page with the 'Add
      Attribute' dialog open; in the attributes table, the 'Options
      ' column header includes a checkbox and an 'Options' button,
      neither focused nor expanded.",
  "action": "click [2372] where [2372] is ",
  "environmental_dynamics": "The 'Options' button in the 'Options'
       column header of the Add Attribute dialog became focused and
       expanded a dropdown list with a single option, 'Select All',
       now visible below the button."
}
```

