# OpenReview forum: "Test-Time Adaptation for LLM Agents via Environment Interaction"
_ICLR.cc/2026/Conference — ICLR 2026 Poster_

### Official Review · Reviewer_oK2P · 2025-10-21

**Soundness:** 2
**Presentation:** 1
**Contribution:** 3
**Rating:** 4
**Confidence:** 4

**Summary:**

This paper proposes 2 test-time adaptation methods for LLM-based agents, which allows the agent explore the environment before the task and update its parameter or context based on the exploration. The experiment shows that this method can improve the performance of the web agent compared with WAM (using world model to predict the next observation and filter the action)

**Strengths:**

1. This idea is natural but effective. The most prior knowledge in the environment agent have, the better the agent can perform.

2. Lifelong learning is also a important topic. This paper try to give a solution to the problem by training. (Although its solution has some weaknesses)

**Weaknesses:**

1. The writing is really poor. Those 2 figures can't really present the idea of 2 methods.

2. The classification of Syntactic Mismatch and Semantic Mismatch is meaningless. They are both the result of the lack of prior knowledge in the environment. Although the author try to seperate them by using 2 different methods, it looks like more weird, since they are both solve the lack of prior knowledge, no matter what prior knowledge it is. Such mapping is useless, it just like to increase the complexity of the paper.

3. Almost half of the results in Table 2 is empty, which may cause unfair comparison. For example, is AWM really worse than your method in other task? (gpt-4o-mini & Qwen). Is it because the AWM's result comes from a old gpt-4o-mini model? You need to reapplied this method in another benchmarks to prove your claim.

4. If the environment can't be explored, this method will not work, although I agree most of the environment can be explored.

As a result, although I quite like this idea, the soundness and the writing are not good enough, so I'll give a borderline score to this paper.

**Questions:**

1. Explain each variation clearly, like I1:n−1 in equation 3.

2. Fully polish your writing, make it more clear and easy to understand.

---

> ### Author Response · Authors · 2025-11-14
> **Inquiry for actionable steps to improve reviewer’s assessment of our work**
>
> We thank you for your thoughtful feedback and are encouraged that you find the core idea and the connection to lifelong learning promising. We will substantially revise the writing, especially the method section and figures, to make the two adaptation procedures and their intuition much clearer.
>
> Regarding the “syntactic” vs. “semantic” mismatch terminology: although both ultimately arise from lack of prior knowledge, we categorize broad observations across different agentic environments (e.g., function-calling, web browsing) into the two failure patterns. We then use these patterns to motivate two adaptation strategies (non-parametric vs. parametric). Our intention is not to introduce a rigid taxonomy, but to provide an operational lens that clarifies why each mechanism addresses a different aspect of missing prior knowledge. We will further refine this explanation in the revision so the distinction feels natural rather than artificial.
>
> While addressing your comments, we would also appreciate a bit of clarification so we can better align the revision with your expectations.
> Regarding your concern on table 2 to be empty:
> - Tau-Bench is a purely conversational benchmark without environment transitions, so environment-dynamics-based methods cannot be applied there. We will make this explicit in the text and table caption and mark these cells as “N/A” rather than leaving them visually empty.
> - WMA cannot be extended to the function-calling benchmark BFCLv3 because it relies on a web-specific data collection and training pipeline. This machinery does not transfer directly to BFCLv3. Our goal in including WMA on WebArena is (i) to compare fairly under its intended setting and (ii) to highlight that our non-parametric methods can generalize across environments where WMA is not directly applicable.
> Regarding the comparison with WMA, to make a fair comparison. We chose GPT 4o-mini, the same model as WMA to compare with it.
>
> **We would also greatly appreciate any guidance on concrete steps that would most improve your assessment of our work**. For example, specific analyses, or clarifications you would consider particularly important. This will help us prioritize the most impactful changes in our paper.

---

> > ### Comment · Reviewer_oK2P · 2025-11-17
> >
> > Thanks for review's response.
> >
> > I believe some of the concern can be only solved based on the writing logic in your paper. So I'll decide whether to raise my socre based on you final version paper. If you have finished updated your paper, please response to this comment!

---

> ### Comment · Reviewer_oK2P · 2025-11-17
>
> mostly, you need to update your figures. It only shows parts of your method and did not reflect any insights behind your method. (just some of the pipeline). I can't get useful imformation if I only read these 2 figure.

---

> > ### Author Response · Authors · 2025-11-26
> >
> > We thank the reviewer for the thoughtful feedback and for recognizing the effectiveness of our core idea regarding lifelong learning in agents. We value the constructive criticism regarding the presentation and definitions, and we have substantially revised the manuscript to address these points.
> >
> > **[W1, Q1 & Q2 Presentation: Figures and Writing]**
> > We have taken the criticism regarding the "Poor" presentation score seriously. We have:
> > - Redesigned Figures 1 & 2 and added explanations in the captions: Instead of partial pipelines, the new figures show the complete end-to-end flow. We added a concrete running example (purchasing on a shopping website) that persists through the figures to ground the abstract concepts.
> > - Polished the Writing: We revised the introduction to clarify that both mismatches and they stem from lack of prior knowledge. We have rewritten the Method section to clarify each notation (including $I_{1:n-1}$) and improved the flow of the Introduction to better motivate the problem setup. We also added additional explanations in table captions for clarity.
> >
> > **[W2 The "Syntactic vs. Semantic" Terminology]** We agree with the reviewer that, at a fundamental level, both failure modes stem from a "lack of prior knowledge." However, we distinguish them operationally because they require fundamentally different solutions:
> > - Syntactic (Format): Requires aligning output distributions (tokens/tool-use structure). This is best solved via Parametric adaptation (gradients), as it requires shifting the model's "parametric memory."
> > - Semantic (Dynamics): Requires reasoning about cause-and-effect. This is best solved via Non-Parametric adaptation (context), as it requires explicit reasoning over new rules about the environments.
> > We have revised the paper to clarify that this taxonomy is a practical framework for selecting adaptation mechanisms, rather than a rigid theoretical division. We have also revised the introduction section to incorporate the reviewer’s comment that they stem from lack of prior knowledge in the environment.
> >
> > **[W3 & W4 Unfair Comparison (Empty Table Cells) & WMA]**
> > *Regarding your concern on table 2 to be empty*:
> > We apologize for the confusion caused by the empty cells. We have updated Table 2 to explicitly mark them as "N/A" and expanded the caption.
> > - **Why WMA is N/A**: WMA is not merely a prompting strategy; it requires training a separate Llama-based world model on environment-specific web trajectories [1].
> >     - Tau-Bench: This is a conversational benchmark with no environment state transitions (no visual observation changes), making a world-model approach mathematically inapplicable.
> >     - BFCLv3: This benchmark lacks the stable web-like state transitions required to train the WMA pipeline.
> > - **Fairness of Comparison**: To ensure the comparison on WebArena was fair, we utilized the exact same backbone model (GPT-4o-mini) for both our method and the WMA baseline [1]. This isolates the gain to the method itself, not the base model.
> >
> > **[Soundness: Environment Exploration]**
> > We agree that if an environment is strictly non-explorable (e.g., a service agent chats with a user and the user is a part of the environment), the non-parametric method has limits. However, most agentic tasks (web, OS, software) allow for our interaction-based exploration. We have added a limitation section acknowledging this boundary condition.
> > We hope these clarifications and the significant revisions to the writing and figures improve your assessment of our work.
> >
> > **References**:
> >
> > [1] Chae, Hyungjoo, et al. "Web agents with world models: Learning and leveraging environment dynamics in web navigation." arXiv preprint arXiv:2410.13232 (2024).

---

> > > ### Comment · Reviewer_oK2P · 2025-11-27
> > >
> > > Thanks for the response. I have raised my score to 6.

---

> > > > ### Author Response · Authors · 2025-11-27
> > > >
> > > > We are glad that we have addressed the reviewer's concerns and we thank the reviewer for raising the score in support our work.

---

### Official Review · Reviewer_uv4G · 2025-10-25

**Soundness:** 3
**Presentation:** 3
**Contribution:** 3
**Rating:** 6
**Confidence:** 4

**Summary:**

This paper addresses the core challenge of Large Language Model (LLM) agents' poor generalization when facing novel, unseen environments. The authors innovatively decompose this challenge into two primary failure modes: Syntactic Mismatch and Semantic Mismatch.

To address these issues, the paper proposes two annotation-free Test-Time Adaptation (TTA) strategies:

1. A parametric (PA) method that learns a lightweight adaptation vector, δ, to bias the model's output distribution, enabling rapid alignment with the environment's syntactic structure.
2. A non-parametric (NPA) method that, prior to task execution, actively learns the environment's causal dynamics through an exploration phase and provides them as context to the agent, thereby constructing a non-parametric world model.

**Strengths:**

1. Introduces Test-Time Adaptation (TTA) into the domain of LLM agents, proposing targeted solutions for syntactic and semantic mismatches.
2. The proposed methods are plug-and-play and exhibit strong adaptability.
3. Provides detailed analysis and comprehensive ablation studies.

**Weaknesses:**

1. The combination of the parametric (PA) and non-parametric (NPA) methods does not demonstrate synergistic effects; in fact, their integration fails to yield better performance.
2. The generalizability of the parametric adaptation method is not sufficiently demonstrated, as its experiments were confined to the Qwen-2.5 family of models.
3. In more diverse and complex environments, the parametric adaptation method might require more training steps, leading to increased computational latency.

**Questions:**

1. Have the authors considered validating the effectiveness of both methods on a broader range of models, such as Llama-3 or GLM-4.5?
2. For the non-parametric adaptation method, the comprehensiveness of the exploration (e.g., exploring the full functionality of a website, not just a subset) is critical. How do the authors ensure that the exploration is sufficiently comprehensive?

---

> ### Author Response · Authors · 2025-12-02
>
> We thank you for your positive assessment of our work’s soundness and contribution. We are glad that you find our plug-n-play adaptation methods to “exhibit strong adaptability” and our studies to be “comprehensive”. We appreciate the constructive questions regarding generalization and exploration coverage. We have conducted additional experiments and analysis to address your concerns.
>
> **[Weakness 1 Synergy of Hybrid Methods]**: You correctly noted that the naive combination of Parametric (PA) and Non-Parametric (NPA) adaptation did not yield synergistic gains. We view this as a significant scientific finding regarding the interference between implicit and explicit adaptation signals. To investigate why, we analyzed episodes where the hybrid method failed but single methods succeeded. Our qualitative analysis revealed two primary interference patterns:
> - **Contextual Overloading**: The model sometimes over-prioritized the explicit dynamics from NPA (e.g., detailed product descriptions) at the expense of the immediate task instruction, leading to "repeating observation" loops.
> - **Signal Conflict**: The gradient updates from PA occasionally drifted the model away from the strict adherence required to follow the NPA rules. This suggests that a naive combination creates conflicting guidance. Future work should focus on a "meta-controller" to gate these methods, rather than applying both simultaneously.
>
> **[Weakness 2 and Question 1 Generalization Across Models]**: We appreciate the suggestion to expand our model suite. While training the 355B GLM-4.5 despite being a MoE model is beyond our computational resources, we conducted new experiments on WebArena using Llama-3.1-8B-Instruct, Ministral-8B-Instruct-2410, and Gemma-3-27B-It to test adaptability across model sizes and families.
>
> | Model| ICL (Baseline) | NPA (Ours) | PA (Ours) |
> |-|-|-|-|
> |Llama-3.1-8B-Instruct| 3.0%|4.0%|3.0%|
> |Ministral-8B-Instruct-2410| 9.0%|8.0%|7.0%|
> |Gemma-3-27B-It| 12.0%|17.0%|N/A*|
> *Note: Due to time constraints during the rebuttal, PA was not run on large Gemma models, but NPA results are provided.
> Here is the analysis:
> - **Scaling with Capability**: We observe that TTA effectiveness correlates with model capability. The larger Gemma-3-27B-It achieved a significant +5% absolute gain (42% relative improvement) using NPA. This aligns with our main paper results (Qwen2.5-14B/32B), confirming that capable models effectively leverage the extracted dynamics.
> - **Limitations of Smaller and weaker Models**: The 8B models (Llama/Ministral) showed marginal or negative changes compared to our Qwen2.5-14B models. This indicates that smaller models may lack the sufficient instruction-following capacity to utilize the complex, extracted world-dynamics (NPA) or the capacity to fine-tune effectively via single-step updates (PA) in zero-shot agentic scenarios. This reinforces our claim that our methods are particularly effective for unlocking the potential of capable, medium-to-large-scale models (e.g., Qwen2.5-14B, GPT-4.1, Gemma3-27B).
>
> **[Weakness 3 Latency of Parametric Adaptation]**: We clarify that PA is highly efficient. In all experiments, we use a single gradient step ($N=1$). As shown in Table 4, the computational overhead is negligible (~3% increase). We validated this further on BFCLv3 multi-turn tasks (which often exceed 20k tokens context). Even at this context length, the latency overhead remained approximately 3%, confirming that PA is suitable for real-time deployment even in complex environments.
> **[Question 2 Comprehensiveness of Exploration]**: We ensure comprehensive exploration through two mechanisms:
> - **Diversity via Personas**: We use distinct personas with specific interests (e.g., "History Buff" vs. "Pet Lover") to drive the agent toward different parts of the state space.
> - **Novelty Heuristic**: During exploration, the agent is explicitly instructed to avoid actions that result in previously observed state transitions.
>
> To validate this empirically, we analyzed 5 exploration logs in the Reddit environment. The personas successfully drove the agent to explore disjoint features of the site:
> - **"History Buff"**: Explored search functionality and image viewing.
> - **"Foodie"**: Explored posting comments, upvoting/downvoting, and the self-edit feature.
> - **"Casual Browser"**: Explored navigation between "sub-reddits" (postmill) and submission creation.
> - **"Pet Lover"**: Explored image zooming and deep-nested comment threads.
> This analysis confirms that our persona-driven approach successfully covers a wide breadth of functionality (search, interactions, account management, navigation) without requiring human intervention.
>
> We hope these additional experiments and analyses address your concerns, and that you will consider raising your assessment.

---

### Official Review · Reviewer_xmMr · 2025-10-31

**Soundness:** 3
**Presentation:** 3
**Contribution:** 3
**Rating:** 6
**Confidence:** 2

**Summary:**

The paper proposes two method for test-time adaptation of LLM-based agents: a "parametric" and a "non-parametric" adaptation. The parametric adaptation proposes, for each episode, to learn a single adaptation vector which is added to the final hidden representation before the final projection layer, i.e., the vector acts as a logit bias. The non-parametric adaptation proposes a one-time deployment pipeline per environment: (1) derive exploration goals from the environment description via LLM calls, (2) run an LLM agent to explore and log (observation, action, new-observation) transitions, (3) have an LLM summarize these logs, (4) have a reasoning model filter out duplicate rules; at test time, the filtered rules are appended to the agent's context.

**Strengths:**

### Clarity

The writing is clear and free of typos. The figures are well designed and effectively illustrate the proposed adaptation strategies. The paper is also well structured: it opens by partitioning LLM-agent failures into two categories and then develops a corresponding adaptation strategy for each, yielding an easy-to-follow narrative.

### Originality

Based on my understanding of the literature, the proposed ideas seem reasonably novel in the context of LLM-based agents.

### Quality

This paper (i) characterizes failure cases of LLM-based agents, (ii) formalizes deployment constraints that bound viable solutions, (iii) derives methods that adhere to those constraints, and (iv) evaluates them against established agent baselines.


### Significance

The paper tackles out-of-distribution (OOD) performance in LLM-based agents which is an essential issue for robust deployment.

**Weaknesses:**

While Section 2 offers useful background on test-time adaptation and LLM-based agents, the paper stops short of situating its specific design choices within the existing literature. For instance, the proposed “parametric adaptation” appears to extend the methodology of “Steering LLM Reasoning Through Bias-Only Adaptation” [1] to the context of LLM-based agents.

[1] Sinii, V., Gorbatovski, A., Cherepanov, A., Shaposhnikov, B., Balagansky, N., & Gavrilov, D. (2025, May 24). Steering LLM reasoning through bias-only adaptation (arXiv:2505.18706 [cs.LG], v1). arXiv. https://arxiv.org/abs/2505.18706

**Questions:**

I strongly suggest to contextualize the design choices for both the “parametric” and “non-parametric” adaptation methods in relation to prior work, highlighting key precedents and  differences.

---

> ### Author Response · Authors · 2025-11-28
>
> We thank you for your thoughtful review. We are encouraged that you found the paper’s writing “clear”, the proposed solution “essential” and “novel” for robust agent deployment. We appreciate your constructive feedback regarding the contextualization of our design choices.
>
> **[Weaknesses & Questions Contextualizing Our Main Methods]**
> We thank the reviewer for highlighting the connection to Sinii et al. (2025) and broader steering vector literature. You are correct that our method shares the structural mechanism of injecting a learned bias into hidden representations. However, we wish to clarify that our Parametric Adaptation is not merely an extension of steering vectors to agents, but rather a distinct online adaptation paradigm tailored to the unique constraints of agentic deployment.
>
> We have revised Section 2 (Related Work) to explicitly contrast our approach with steering methods and discuss connections between them. Specifically, we added the following distinctions:
>
> - **Target of Adaptation (High-level Traits vs. Local Syntax)**: Standard steering approaches [1,2,3,4] typically apply vectors to shift global, high-level traits such as "honesty" or "reasoning style." In contrast, our approach uses the vector to align the model with local, environment-specific syntax (e.g., unique observation formats or API schemas) that cannot be defined a priori.
> - **Optimization Paradigm (Static vs. Online)**: Steering vectors are generally pre-computed offline [1,3,4] or treated as fixed modulations during inference [2]. Our method treats the vector as a temporary parameter that is optimized online via gradient descent to adapt to the environment dynamically, using the environment’s immediate response as a self-supervisory signal.
> - **Lifecycle (Global vs. Episodic)**: Unlike most steering vectors which are usually applied globally across inputs [1,2,3,4], our adaptation is reset per episode. This ensures the model adapts to the specific dynamics of the current task without suffering from interference across different environments.
>
> In summary, while the mechanism is similar, our contribution lies in repurposing this mechanism for rapid, episodic, online distributional alignment rather than static behavioral steering.
>
> **Contextualizing Non-Parametric Adaptation**:
> We appreciate the suggestion to better situate our Non-Parametric Adaptation within existing literature. In the revised manuscript, we have strengthened the distinction between our method and prior art:
>
> - **vs. World Models (e.g., WMA)**: Unlike heavy-weight approaches that require collecting data and training a separate neural network to predict state transitions (which is computationally expensive and data-hungry) [5], our Non-Parametric Adaptation builds a lightweight, text-based world model "in-context."
> - **vs. RAG/Memory (e.g., AWM)**: While methods like Agent Workflow Memory [6] extract from past successful trajectories, our method is distinct and conceptually complementary because it performs active, persona-driven exploration to discover environment dynamics before a task is even attempted, or trajectory is verified. This allows the agent to handle "cold start" scenarios where no successful trajectories yet exist.
>
> We believe these clarifications regarding the distinct "online and episodic" nature of our parametric method and the "active exploration" nature of our non-parametric method directly address your concerns about contextualization. We have integrated these comparisons into the final paper to ensure the design choices are clearly justified. We hope you would consider raising your score in support of our work.
>
> **References**:
> [1] Tan, Daniel, et al. "Analysing the generalisation and reliability of steering vectors." Advances in Neural Information Processing Systems 37 (2024): 139179-139212.
> [2] Subramani, Nishant, Nivedita Suresh, and Matthew E. Peters. "Extracting latent steering vectors from pretrained language models." arXiv preprint arXiv:2205.05124 (2022).
> [3] Sinii, Viacheslav, et al. "Steering LLM Reasoning Through Bias-Only Adaptation." arXiv preprint arXiv:2505.18706 (2025).
> [4] Rimsky, Nina, et al. "Steering llama 2 via contrastive activation addition." Proceedings of the 62nd Annual Meeting of the Association for Computational Linguistics (Volume 1: Long Papers). 2024.
> [5] Chae, Hyungjoo, et al. "Web agents with world models: Learning and leveraging environment dynamics in web navigation." arXiv preprint arXiv:2410.13232 (2024).
> [6] Wang, Zora Zhiruo, et al. "Agent workflow memory." arXiv preprint arXiv:2409.07429 (2024).

---

### Author Response · Authors · 2025-12-02
**Author Final Remarks**

We thank the reviewers for their engagement and appreciate their positive feedback: our proposed methods were described as `"novel"` [xmMr], `"essential"` for robust deployment [xmMr], `"effective"` [oK2P], and `"plug-and-play"` exhibiting `"strong adaptability"` [uv4G]. Reviewers also highlighted our `"comprehensive ablation studies"` [uv4G] and noted that the paper tackles the critical issue of OOD performance in agents [xmMr].

We appreciate the thoughtful feedback and concerns raised by the reviewers. In response, we have carefully addressed each point by providing additional experimental results, extensively revising the manuscript, and clarifying design choices. Below, we summarize the main concerns from the reviewers and outline how we have addressed them.
- For **reviewer oK2P**, we addressed concerns regarding the presentation quality and the meaningfulness of our failure mode taxonomy. We **substantially redesigned Figures 1 & 2** to show complete end-to-end flows with running examples and **polished the writing** to clarify definitions. We also clarified that our "Syntactic vs. Semantic" distinction is an operational framework for selecting adaptation mechanisms (gradients vs. context) rather than a rigid taxonomy. Furthermore, we updated Table 2 to explicitly mark N/A cases for the WMA baseline to ensure fair comparison. **We are particularly grateful that reviewer oK2P acknowledged these revisions and the effectiveness of our method, commented to raise the score to 6**.
- In response to **reviewer xmMr**, we focused on **contextualizing our work** within existing literature, specifically regarding steering vectors. We revised Section 2 to explicitly contrast our Parametric Adaptation with steering vectors, highlighting three key distinctions: 1) Target: We align with local syntax rather than high-level traits; 2) Optimization: We utilize online, self-supervised gradient descent rather than pre-computed static vectors; and 3) Lifecycle: Our adaptation is episodic to prevent interference. We also clarified the relationship between our Non-Parametric Adaptation and existing World Model/RAG approaches.
- For **reviewer uv4G**, we addressed questions regarding generalization, hybrid method synergy, and exploration coverage. We **conducted new experiments** with a broader suite of models (Llama-3.1-8B-Instruct, Ministral-8B-Instruct-2410, and Gemma-3-27B-It), demonstrating that our adaptation effectiveness scales with model capability (e.g., +5% absolute gain for Gemma-3-27B-It). We **provided a qualitative analysis explaining that the lack of synergy in the hybrid approach** stems from **signal interference** between implicit gradients and explicit context. Additionally, we **validated the efficiency** of our parametric method (~3% latency overhead) and **provided empirical evidence** from Reddit logs showing that our persona-driven exploration successfully covers diverse environment features.

We sincerely thank the AC for navigating this difficult review cycle and taking on the substantial, delegated effort required to render a final decision under the constraints caused by the data leak.

---

### Meta-Review · Area_Chair_PgUZ · 2026-01-02

**Summary:**

This paper addressed the problem of poor generalization of LLM-based agents in novel environments. Specifically, the authors proposed two annotation-free Test-Time Adaptation strategies that allow agents to explore and adjust prior to task execution. The first strategy is a parametric method that learns a lightweight adaptation vector to rapidly align the model's output distribution. The second is a non-parametric method by actively exploring the environment to learn causal dynamics and summarizing them into rules that are appended to the context. Experiments demonstrate that the proposed methods significantly improve web agent performance compared to baselines.

**Reviewer Concerns:**

One reviewer raised concerns on the writing of the paper (including figures). The authors successfully revised the paper to reduce this concern (and finally the reviewer was satisfied). Some reviewers raised the concerns on the scalability and generalizability of the proposed method and the authors provided additional results during the rebuttal to reduce the concerns.

**Reviewer Scores:**

Initial score was 6,6,4. After the discussion, the last reviewer raised the score from 4 to 6. The final scores are 6, 6, 6. There are no major concerns that are not resolved during the rebuttal period.

---

### Decision · Program_Chairs · 2026-01-26

Accept (Poster)